# Effects of Lead Pollution on Photosynthetic Characteristics and Chlorophyll Fluorescence Parameters of Different Populations of *Miscanthus floridulus*

**Jianqiao Qin** [1], **Xueding Jiang** [2,*], **Jianhua Qin** [3], **Huarong Zhao** [4,*], **Min Dai** [1], **Hao Liu** [1] and **Xi Chen** [5]

[1] Guangdong Provincial Key Laboratory of Environmental Health and Land Resource, School of Environmental and Chemical Engineering, Zhaoqing University, Zhaoqing 526061, China; qinjianqiaosci@126.com (J.Q.); daimin1007@163.com (M.D.); liuhao13189340196@126.com (H.L.)

[2] School of Environmental and Chemical Engineering, Foshan University, Foshan 528000, China

[3] School of Foreign Studies, South China Agricultural University, Guangzhou 510640, China; jackyqin04@scau.edu.cn

[4] College of Environmental Science and Engineering, Guilin University of Technology, Guilin 541004, China

[5] School of Environmental Science and Engineering, Sun Yat-sen University, Guangzhou 510275, China; chenxi90511@163.com

[*] Correspondence: jiangxueding@fosu.edu.cn (X.J.); zhaohuar@mail3.sysu.edu.cn (H.Z.)

**Abstract:** This study was conducted in order to study the effect of different concentrations of lead pollution on the photosynthetic characteristics and growth of *Miscanthus floridulus*, and to reveal its photosynthetic adaptability to lead stress. The differences of gas exchange parameters, chlorophyll fluorescence characteristics and photosynthetic pigment of two *Miscanthus floridulus* populations, one population from Boluo an uncontaminated site, and another population from Dabaoshan, a mine site, were compared and studied through nutrient solution culture experiments treated with heavy metal lead (Pb) in green house. The results showed that (1) under Pb stress, the net photosynthetic rate (Pn), transpiration rate (Tr), stomatal conductance (Gs), intercellular carbon dioxide concentration (Ci), and chlorophyll content (Chl) of the leaves of the two populations decreased in different amplitude. Under moderate and severe Pb stress (80 mg·L$^{-1}$, 120 mg·L$^{-1}$, 240 mg·L$^{-1}$), the plant biomass of non-mining population and mining population plants were 54.5%, 39.7%, 29.4% and 70.4%, 54.7%, 50.9% of the control, respectively. (2) Stomatal restriction was the main factor for the Pn's decrease in the leaves of the non-mine population under light Pb stress, while the non-stomatal restriction was the main factor for Pn's decrease in the leaves of the non-mine population under middle and high Pb stresses. (3) Under Pb stress, the maximum photochemical efficiency (Fv/Fm) and potential activity (Fv/Fo) of PS II reaction centers in the two populations of *M. floridulus* decreased. However, Fv/Fm and Fv/Fo showed a smaller decrease, but the capability to utilize light and the potential to activate PSII of the mine population remained higher than that of the non-mine population. The changes of photochemical quenching coefficient (qP) and non-photochemical quenching coefficient (NPQ) of PSII showed that qP value decreased and NPQ value increased in the two populations under Pb stress. On the whole, the resistance mining area population had a low qP reduction and a large increase in NPQ. Electronic transfer rate (ETR) and PSII actual fluorescence efficiency (ΦPSII) of the mine population changed slightly under Pb stress. These results indicated that when under Pb stress, the electron transport activity and photosynthetic apparatus were damaged less in the mine population with high resistance than in the non-mine population with low resistance. Conclusion: the mining population of *M. floridulus* has strong tolerance to Pb, which is suitable for the pioneer species of gramineae in vegetation restoration construction in metal mining area.

**Keywords:** lead pollution; *Miscanthus floridulus*; gas exchange parameter; chlorophyll fluorescence

## 1. Introduction

Pb is one of the most serious environmental heavy metal pollutants [1]. Excessive amounts of heavy metals in soil on plant is a kind of coercion or adversity condition; if the stress on the plant itself is limited, it will damage the plant [2]. The toxicology of lead stress is the nucleus, mitochondria, and chloroplast ultrastructure destruction of plants, which reduces chlorophyll and ascorbic acid content, reduces dehydrogenase activity of nitrate reductase, hinders the plant respiration metabolism, photosynthesis, hydrogen reduction, cell division normal physiological function and, ultimately, affects the quality of biomass plants [3,4]. According to statistics, at present, the cultivated land area polluted by heavy metals in China is almost $2.0 \times 10^7$ hm$^2$, accounting for about 1/5 of the cultivated land area. The direct economic loss of grain caused by heavy metal pollution in China is worth tens of billions of yuan every year [5]. These harms to the ecosystem attracted the attention of researchers [6]. Phytoremediation can remove heavy metals using plants and offer the benefits of low cost, as well as being an environmentally sustainable technique [7]. To date, about 700 species of plants were reported to be hyperaccumulators of different contaminants [8]. Metal hyperaccumulators are found in a large number of plant families, but most are found in the Brassicaceae family [9]. Moreover, hyperaccumulators are associated with slow plant growth and low biomass yields, and so, there is an urgent need for identification of other plant species having fast growth and greater biomass production [10]. Phytoremediation would be ideal for the recovery of various types of contaminated soil, since it is effective and environmentally friendly [11].

The *Miscanthus floridulus* is a C4 perennial grass that originates from East Asia, which is widely used as forage and ornamental grass, and it is cultivated as bioenergy crop for lignocellulosic ethanol production, owing to high biomass productivity potential, high efficient photosynthesis, water use efficiency, and great resistance to drought, salt, and heavy metal stress [12,13]. *M. floridulus* is commonly planted in vast marginal land, with limited water supply and heavy metal pollution [14]. It is an ideal plant for phytoremediation due to its fast growth, large biomass, and well-developed roots system, which can accumulate heavy metals in the roots and reduce the mobility and availability [15,16]. Because of its long-term survival in the polluted environment, *M. floridulus* from the metal mining area may have undergone resistance evolution, forming a resistant ecotype [17]. Facultative metal-type plants growing in heavy metal contaminated areas are generally genetically different from plants of the same species in non-contaminated areas [18]. Metalliferous plants are of particular interest for the study of plant anti-pollution mechanisms and phytoremediation of heavy metal contaminated environments. Therefore, identifying their growth limiting factors is of great significance for the practical application of phytoremediation technology [19].

The Dabaoshan mine, located at the junction of Qujiang County and Wengyuan County, Shaoguan City, Guangdong Province, is a large iron polymetallic sulfide-associated deposit [20]. Zhao et al. [21] investigated the soil of Dabaoshan and found that more than 57.98% and 55.09% of the soils in the study area showed Zn and Pb pollution. According to a study of plants in the Dabaoshan mining area by Chen et al. [22], the rhizosphere of *M. floridulus* can activate Pb and other heavy metals significantly and may be considered a pioneer plant for local vegetation restoration. Previous studies also showed that *M. floridulus* can grow normally in the heavily polluted tailings ponds in the Dabaoshan mining area and that its roots can absorb and fix heavy metal Pb [23,24]. Moreover, its biomass is large, and so, it represents good plant repair material [25,26].

Due to the long-term survival in the polluted environment, the resistance evolution in the metal mining area may have occurred in accordance with the polluted environment [16], forming the resistance ecotype. Studying the physiological and ecological differences between resistant ecotypes and sensitive ecotypes is an important way to reveal the mechanism of plant resistance and make phytoremediation technology of heavy metal pollution widely applied [27,28]. At present, relevant studies focus on heavy metal resistance and its accumulation performance [29], while there are few studies on leaf photosynthesis and source-pool allocation of photosynthetic products, which are closely related to plant growth [30].

In this paper, the photosynthetic characteristics and chlorophyll fluorescence parameters of two populations of *M. floridulus* from Dabaoshan (mining area) in Shaoguan City, Guangdong Province, and Boluo County (non-mining area) in Huizhou City, Guangdong Province, were compared through the nutrient solution culture experiment, so as to explore the differences in the physiological and ecological responses of *M. floridulus* populations to Pb pollution in polluted and non-polluted areas. Our aim was to further explore the mechanism of ecological adaptation of Pb resistance in plants. At the same time, the vegetation restoration of tailings and abandoned land is not only faced with heavy metal pollution, but also faced with environmental pressure caused by poor soil water retention and drought. The physiological and ecological characteristics of net photosynthetic rate, transpiration rate, and water use efficiency of *M. floridulus* were studied in order to provide theoretical basis for photosynthetic physiological and ecological aspects of vegetation restoration in abandoned heavy metal mining area.

## 2. Materials and Methods

### 2.1. Test Materials

The seeds of the mining population were collected from Dabaoshan mining area in Shaoguan, Guangdong (latitude and longitude: 24°33′36.6″ N, 113°43′14.0″ E), and the seeds of the non-mining population were collected from Boluo County, Huizhou, Guangdong (latitude and longitude: 24°33′36.6″ N) (23°08′57.8″ N, 114°21′10.0″ E). All belonged to the same subtropical monsoon climate region. The soil properties of the seed sources are shown in Table 1.

**Table 1.** The basic chemical properties of the local soils supporting the two natural populations of *M. floridulus*.

| Soil Sample Point | pH | Organic C (g·kg$^{-1}$) | Available P (mg·kg$^{-1}$) | Available N (mg·kg$^{-1}$) | Heavy Metal Contents (mg·kg$^{-1}$) | | | |
|---|---|---|---|---|---|---|---|---|
| | | | | | Zn | Pb | Cu | Cd |
| Dabaoshan Mining Area | 6.2 | $14.7 \pm 0.9$ b | $32.2 \pm 2.0$ b | $30.2 \pm 1.9$ b | $1768.7 \pm 91.1$ a | $1253.3 \pm 71.3$ a | $1701.3 \pm 77.5$ a | $9.1 \pm 0.9$ a |
| Boluo County | 6.6 | $13.8 \pm 0.9$ b | $26.6 \pm 1.8$ b | $28.4 \pm 2.7$ b | $135.2 \pm 13.1$ b | $242.6 \pm 44.1$ b | $48.4 \pm 9.5$ b | $1.1 \pm 0.2$ b |

Note: Data in the table are means $\pm$ SD ($n = 3$), different letters in the same vertical column indicate significant difference according to SSR test ($p < 0.05$), the same below.

### 2.2. Experimental Design

Seeds were germinated in nutrient-rich soil, and after 90 days of growth, plants with favorable and consistent growth were selected, washed with tap water, and moved to Hogland nutrient solution for culture. The nutrient solution formula was as follows (mmol·L$^{-1}$): Ca (NO$_3$)$_2$·4H$_2$O 0.625 + KH$_2$PO$_4$ 0.025 + MgSO$_4$ 0.25 + KNO$_3$ 0.625 + (μmol·L$^{-1}$) H$_3$BO$_3$ 2.50 + MnCl$_2$·4H$_2$O 0.5 + ZnSO$_4$·7H$_2$O 0.5 + CuSO$_4$·5H$_2$O 0.05 + (NH$_4$) 6Mo$_7$O$_{24}$ 0.005 + Fe-EDTA 0.125 [31]. After seven days of pre-culture in a greenhouse, the treatment was performed. Single Pb setting involved 6 treatments (mg·L$^{-1}$): CK(No Pb added), 20, 40, 80, 120, and 240; Pb was added to the nutrient solution with Pb (NO$_3$)$_2$. Reduce the concentration of KH$_2$PO$_4$ in all nutrient solutions to 0.005 mmol·L$^{-1}$ to avoid precipitation. There were three replicas in each treatment and three saplings in each replica. The nutrient solution was changed every 4 days, and the pH was adjusted to about 5.8 with 0.1 mol·L$^{-1}$ HCl or NaOH. The physiological index was determined after 40 days of culture.

### 2.3. Determination of Gas Exchange Parameters

Net photosynthetic rate (Pn), transpiration rate (Tr), stomatal conductance (Gs), and intercellular CO$_2$ concentration (Ci) were measured by LI-6800 portable photosynthesis measuring instrument (Li-6800, LI-COR Bioscience, Lincoln, NE, USA). The measurement conditions were: CO$_2$ concentration at 380 μmol·mol$^{-1}$, temperature at 27 °C, relative humidity at 60%, and photosynthetically active radiation (PAR) at 600 μmol m$^{-2}$·s$^{-1}$. Select healthy and fully unfolded leaves from the upper part of each treated plant for measurement. For

each treatment, measure 3 plants and take the average value. Stomatal limit value (Ls) and instantaneous leaf water use efficiency (WUE) were calculated by a formula, namely Ls = 1 − Ci/Ca [32] (Ca is atmospheric $CO_2$ concentration). WUE = Pn/Tr [33].

### 2.4. Determination of Chlorophyll Fluorescence Parameters

The kinetic parameters of chlorophyll fluorescence induction were determined by a portable pulse-modulated chlorophyll fluorescence analyzer (PAM-2000, Walz, Effeltrich, Germany). According to Ralph's [34] method, the leaves were dark adapted for 30 min, and initial fluorescence ($F_0$), maximum fluorescence ($F_m$), steady-state fluorescence ($F_s$), maximum fluorescence ($F_m'$), and minimum fluorescence ($F_0'$) were measured. The photochemical fluorescence quenching coefficient (qP) and non-photochemical quenching coefficient (NPQ) were measured. Based on the above data: variable fluorescence Fv (Fv = $F_m$ − $F_0$), maximum photochemical efficiency of PSII (Fv/Fm), potential activity of PSII (Fv/$F_0$), energy capture efficiency of PSII (Fv'/Fm'), PSII effective photochemical quantum yield ($\phi$PSII = (Fm' − Fs)/Fm'), and the electron transfer rate ETR (ETR = ɸPSII × PAR × 0.5 × 0.84) [35,36].

### 2.5. Determination of Plant Morphology

Select plants with consistent growth, and measure their height with a scale (0.1 cm); the electronic balance (0.0001 g) was used to measure the biomass of the dried underground and overground parts, and calculate the root shoot ratio; in addition, select the third leaf from top to bottom on the top of the plant, measure the leaf area with the leaf area meter LI-3000A, and then measure the dry weight of the leaf with the electronic balance (0.0001 g). Calculate the specific leaf area: specific leaf area (SLA) = leaf area/dry weight of the leaf [36].

### 2.6. Determination of Photosynthetic Pigment Content

Using a 752 N type ultraviolet spectrophotometer, take leaves at the same position and maturity as measured for photosynthesis, and use the extraction method to determine photosynthetic pigments (chlorophyll a, chlorophyll b, and carotenoids) [36]. Measurements were performed on 3 plants for each treatment and repeated 3 times.

### 2.7. Data Processing

The statistical analysis of data was performed using a combination of Microsoft Excel 2007 and IBM SPSS Statistics 22.0 software. Two-way statistical analysis (two-way ANOVA) was used to examine the significant difference between plant and lead treatment. The significance of differences between means was analyzed using Duncan's multiple comparisons test (SSR test, $p < 0.05$).

## 3. Results

### 3.1. Growth Morphological Changes of M. floridulus under Pb Stress

After 15 days of Pb treatment, both populations grew strongly, with fresh roots and leaves. After 30 days, the population in the mining area still grew well, and new leaves continued to grow, while the low concentration treatment group of the non-mining area population also grew well, but the plants in the 120 mg·L$^{-1}$ and 240 mg·L$^{-1}$ treatment groups began to have the symptoms of leaf chlorosis and wilting.

The changes in plant height and biomass of *M. floridulus* are shown in Table 2. As the degree of Pb stress increased, both populations showed a downward trend. Under 80 mg·L$^{-1}$, 120 mg·L$^{-1}$, and 240 mg·L$^{-1}$ Pb stress, the plant height of non-mining population and mining population plants were 48.7%, 39.4%, 31.6% and 71.1%, 56.6%, 51.1% of the control, respectively. Evidently, the plant height of mining population plants decreased less under higher Pb treatment, while that of non-mining population plants decreased significantly. Under moderate and severe Pb stress (80 mg·L$^{-1}$, 120 mg·L$^{-1}$, 240 mg·L$^{-1}$), the plant biomass of non-mining population and mining population plants were 54.5%, 39.7%, 29.4% and 70.4%, 54.7%, 50.9% of the control, respectively. Evidently, the biomass

of mining population plants decreased less under higher Pb treatment, while that of non-mining population plants decreased significantly. In particular, the plant biomass of mining population plants was 1.72 times higher than that of non-mining population plants treated with 240 mg·L$^{-1}$.

**Table 2.** Growth of two *M. floridulus* populations under different Pb treatment.

| Population | Pb Treatment (mg·L$^{-1}$) | Plant Height (cm) | Plant Biomass (g) | Specific Leaf Area (cm$^2$·mg$^{-1}$) | Root-Shoot Ratio |
|---|---|---|---|---|---|
| Non-mining population | 0 | 41.1 ± 3.15 ab | 4.25 ± 0.20 a | 193.3 ± 3.1 a | 0.40 ± 0.07 ef |
| | 20 | 32.1 ± 1.00 c | 3.28 ± 0.02 bc | 181.8 ± 8.1 ab | 0.41 ± 0.04 def |
| | 40 | 26.0 ± 1.45 de | 2.62 ± 0.34 de | 172.3 ± 9.1 bc | 0.45 ± 0.03 cde |
| | 80 | 20.1 ± 2.15 f | 2.32 ± 0.04 e | 152.5 ± 9.5 d | 0.49 ± 0.08 bgd |
| | 120 | 16.2 ± 1.76 g | 1.69 ± 0.42 fg | 143.4 ± 20.2 d | 0.55 ± 0.03 ab |
| | 240 | 13.0 ± 1.95 g | 1.25 ± 0.19 g | 113.4 ± 10.6 e | 0.61 ± 0.03 a |
| Mining population | 0 | 40.8 ± 1.85 ab | 4.22 ± 0.31 a | 191.8 ± 8.2 a | 0.39 ± 0.04 ef |
| | 20 | 42.0 ± 1.95 a | 4.32 ± 0.22 a | 196.7 ± 5.0 a | 0.37 ± 0.02 f |
| | 40 | 38.1 ± 1.45 b | 3.65 ± 0.59 b | 183.7 ± 9.1 ab | 0.42 ± 0.01 def |
| | 80 | 29.0 ± 1.95 cd | 2.97 ± 0.27 cd | 172.1 ± 9.9 bc | 0.45 ± 0.04 def |
| | 120 | 23.1 ± 1.00 ef | 2.31 ± 0.20 e | 162.3 ± 9.6 cd | 0.48 ± 0.08 bcd |
| | 240 | 21.0 ± 2.55 f | 2.15 ± 0.07 ef | 145.5 ± 14.2 d | 0.51 ± 0.02 bc |

Note: Data in the table are means ± SD (*n* = 3); different letters in the same vertical column indicate significant difference according to SSR (New Complex Range Method) test (*p* < 0.05), the same below.

With the increase in Pb treatment concentration, the specific leaf area of different populations of *M. floridulus* showed an overall trend of decline (see Table 2). Under 80 mg·L$^{-1}$, 120 mg·L$^{-1}$ and 240 mg·L$^{-1}$ Pb stress, the specific leaf area of non-mining population and mining population plants were 78.7%, 74.4%, 58.5% and 89.6%, 84.5%, 75.9% of the control, respectively. Evidently, the specific leaf area of mining populations decreased relatively slight under higher Pb treatment, while that of non-mining populations decreased significantly (*p* < 0.05). As can be seen from Table 2, the root/shoot ratio of *M. floridulus* of different populations showed an overall upward trend with the increase in Pb stress concentration, but there were significant differences in the root/shoot ratio under high concentrations (120 mg·L$^{-1}$ and 240 mg·L$^{-1}$) stress: the root/shoot ratios of mining and non-mining populations were 123.1%, 131.2% and 137.5%, 152.5%, respectively, which were significantly different from the control (*p* < 0.05).

*3.2. The Changes of Photosynthesis Indexes*

3.2.1. Effects of Pb on Net Photosynthetic Rate (Pn) and Stomatal Conductance (Gs) in Leaves of *M. floridulus*

Photosynthetic rate is the most important indicator to reflect the strength of photosynthesis. A high photosynthetic rate indicates a high level of plant photosynthesis, and vice versa [37]. As can be seen from Figure 1A, there were obvious differences in the net photosynthetic rate of the leaves of mining population and non-mining population: the populations from the mining area (Dabaoshan) had a minor effect on the net photosynthetic rate of leaves under the stress of different concentrations of Pb, while the populations from non-mining area (Boluo) significantly decreased with the increase in treatment concentration. Specifically, with the increase in Pb stress concentration, the leaf Pn value of both populations decreased, and under 40 mg·L$^{-1}$, 80 mg·L$^{-1}$, and 160 mg·L$^{-1}$ Pb stress, the Pn values of the mining population and non-mining population were 80.8%, 73.3%, 62.1% and 51.9%, 43.2%, 27.5% of the control population, respectively. Evidently, the decrease in the net photosynthetic rate of the mining population was relatively small under the higher Pb treatment, while the decrease in the non-mining population was significant, especially under the 240 mg·L$^{-1}$ treatment. The net photosynthetic rate of the mining population was 2.17 times that of the non-mining population. Through regression analysis, for non-mining population Pn = −0.035x + 7.56 (x is the concentration of Pb, $R^2$ = 0.68, *p* < 0.05), mining

population Pn = −0.021x + 9.15 (x is the concentration of Pb, $R^2$ = 0.82, $p$ < 0.05), The results indicated that heavy metal Pb had a great effect on leaf Pn of the *M. floridulus* population in the non-mining area.

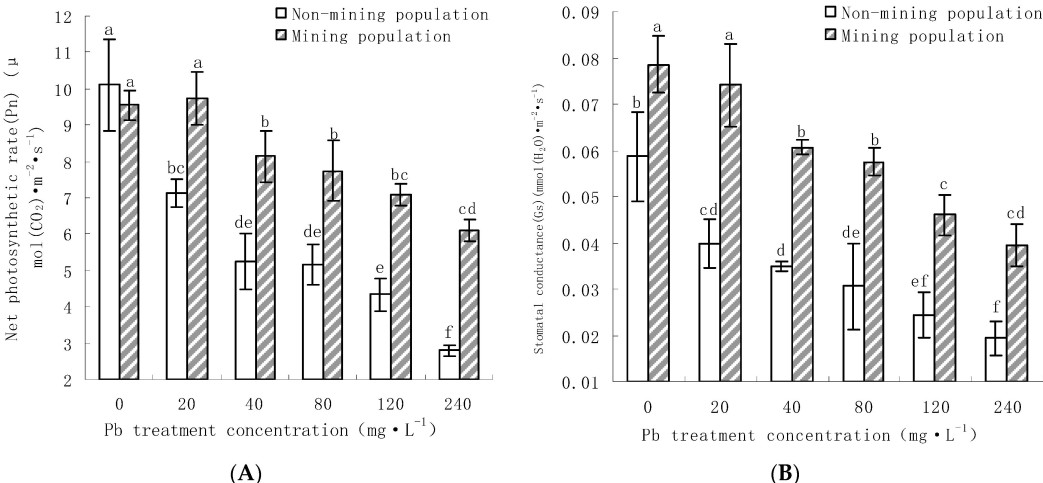

**Figure 1.** Net photosynthetic rate (Pn) and Stomatal conductance (Gs) of two *Miscanthus floridulus* populations under different Pb treatment((**A**) is Pn, (**B**) is Gs). Note: Error bars indicate standard deviation; Different letters in the same group indicate significant difference at $p$ < 0.05 according to Duncan's multiple range tests; the same below.

The effect of Pb on the stomatal conductance of the leaves of the two populations of *M. floridulus* is shown in Figure 2A. Both populations showed a decreasing trend with the increase in Pb concentration, and the non-mining populations under moderate and severe (120 mg·L$^{-1}$ and 240 mg·L$^{-1}$) Pb stress were 40.7% and 31.8% of the control populations, respectively. The mining population was 59.3% and 50.5% of the control, respectively, indicating that the decline of the non-mining population was greater than that of the mining population.

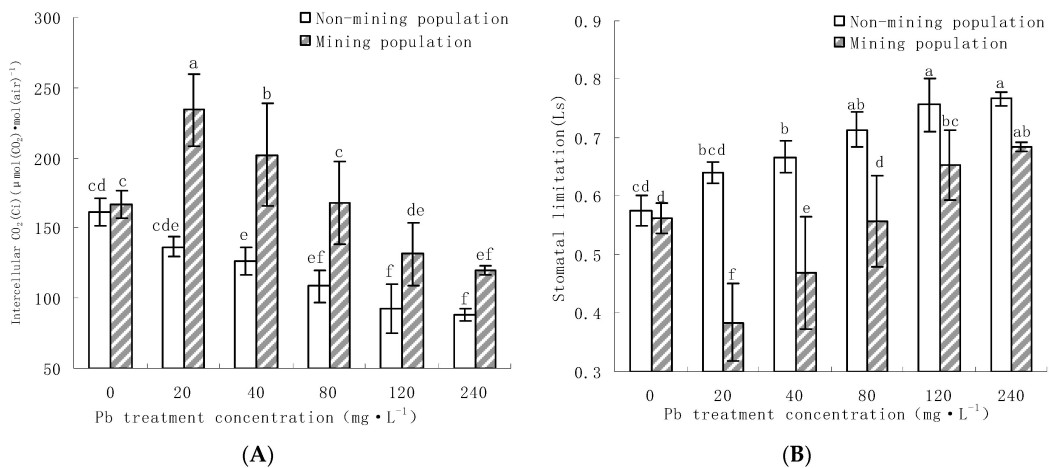

**Figure 2.** Intercellular $CO_2$ concentration (Ci) and stomatal limitation (Ls) of two *Miscanthus floridulus* populations under different Pb treatment ((**A**) is Ci, (**B**) is Ls). Note: Error bars indicate standard deviation; Different letters in the same group indicate significant difference at $p$ < 0.05 according to Duncan's multiple range tests; the same below.

### 3.2.2. Effects of Pb on Intercellular Carbon Dioxide Concentration (Ci) and Stomatal Limit (Ls) in Leaves of *M. floridulus*

As can be seen from Figure 2B, the intercellular carbon dioxide concentration (Ci) of mining population and non-mining population showed different trends under the treatment of different concentrations of Pb: the non-mining population decreased with the increase in treatment concentration, and the Ci value decreased significantly under 120 mg·L$^{-1}$ and 240 mg·L$^{-1}$ Pb stress, which was 57.1% and 53.5% of the control, respectively. The Ci value of mining population increased significantly at the low concentration (40 mg·L$^{-1}$ and 80 mg·L$^{-1}$) treatment ($p < 0.05$), and decreased to a certain extent at the intermediate concentration and elevated concentration treatment. When the concentration of Pb was 240 mg·L$^{-1}$, the Ci value was 73.2% of the control. The results show that the effect of Pb stress on Ci was much larger in the non-mining population than in the mining population.

Stomatal limitations (Ls) were varied by intercellular $CO_2$ and atmospheric $CO_2$ concentration. The stomatal limit of the non-mining population increased significantly with the increase in Pb concentration (Figure 3B), while the mining population decreased at low concentration stress and increased significantly at extreme concentration (120 mg·L$^{-1}$ and 240 mg·L$^{-1}$). Clearly, under Pb stress, the intracellular limit value of the non-mining population was significantly larger than that of the mining population. The analysis of variance showed that there was a significant difference between Ls and control under high Pb stress ($p < 0.05$).

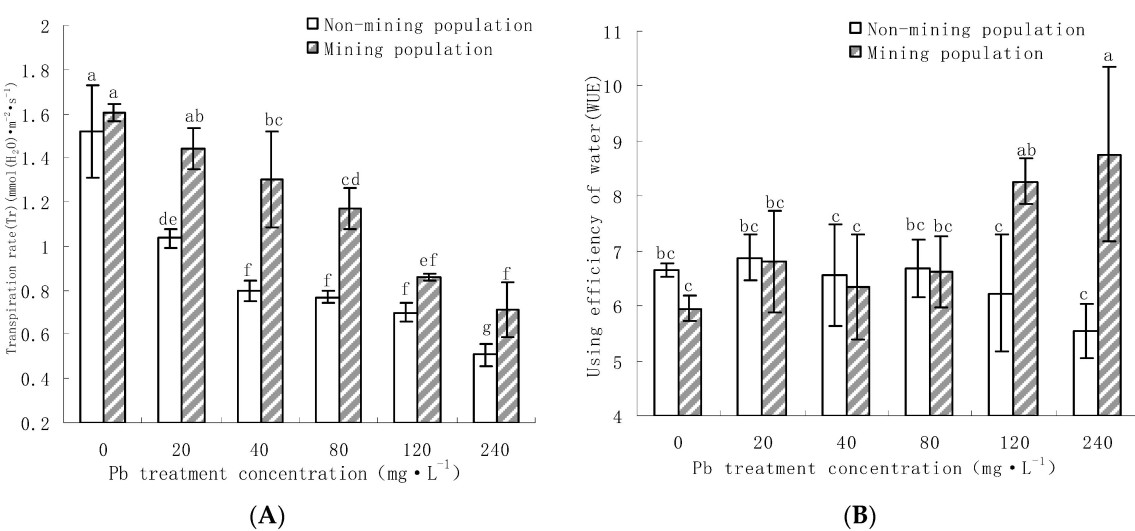

**Figure 3.** Transpiration rate (Tr) and using efficiency (mmolm$^{-2}$·s$^{-1}$) of water (WUE) in two *Miscanthus floridulus* populations under different Pb treatment ((**A**) is Tr, (**B**) is WUE). Note: Error bars indicate standard deviation; Different letters in the same group indicate significant difference at $p < 0.05$ according to Duncan's multiple range tests; the same below.

### 3.2.3. Effects of Pb on Transpiration Rate (Tr) and Water Use Efficiency (WUE) in Leaves of *M. floridulus*

Transpiration is the main driving force for plants to absorb and transport water. It can maintain the water saturation of various parts of plants, maintain the morphology of cell tissues, promote the distribution of inorganic salts in plants, and dissipate the excess heat generated by plants during photosynthesis and oxidative metabolism [38]. The variation of the transpiration rate is shown in Figure 3A. With increasing Pb stress, the Tr value decreased significantly in both populations, but the TR value decreased in different treatments. Under mild Pb stress (20 mg·L$^{-1}$, 40 mg·L$^{-1}$), the Tr value of the non-mining population decreased significantly compared with the control population, but the Tr value of the mining population did not decrease significantly. Under moderate and severe Pb stress (120 mg·L$^{-1}$, 240 mg·L$^{-1}$), the Tr value of non-mining population decreased significantly to 46.6% and 33.3%, the Tr

value of the mining area population decreased to 52.9% and 48.6%. Overall, the transpiration rate of the mining population was significantly higher than that of the non-mining population at different Pb concentration stresses.

The value of water use efficiency (WUE) is directly determined by the transpiration rate and net photosynthetic rate [39]. As can be seen from Figure 3B, there was no significant difference in water use efficiency between non-mining population and mining population under control and low concentration Pb (20 mg·L$^{-1}$ and 40 mg·L$^{-1}$) treatment ($p > 0.05$), while high concentration Pb (120 mg·L$^{-1}$ and 240 mg·L$^{-1}$) treatment, the population in mining area was significantly higher than that in non-mining area ($p < 0.05$). There were also different trends in the variation of the population in mining and non-mining areas. The water use efficiency of non-mining area population increased at low concentration treatment, but decreased sharply with the increase in treatment concentration, and it reached the lowest at 240 mg·L$^{-1}$ treatment. The population water use efficiency increased with the increase in treatment level, and reached the highest level at 240 mg·L$^{-1}$, which was significantly different from that of the control ($p < 0.05$), and was 148.8% of that of the control.

### 3.3. Changes of Chlorophyll Fluorescence Parameters

3.3.1. Effects of Pb on Fv/Fm and Fv/F$_0$ of the Leaves of *M. floridulus*

Fv/Fm reflects the maximum photochemical efficiency of PSII, and Fv/F$_0$ reflects the potential activity of PSII, both of which are essential parameters to indicate the status of photochemical reactions [40].

The results showed that with the increase in Pb stress concentration, both Fv/Fm and Fv/F$_0$ decreased (Figure 4), but the decrease amplitude of the two populations was different. Compared with the control, the decrease in Fv/Fm and Fv/F$_0$ in the two populations under severe Pb stress (240 mg·L$^{-1}$) was 3.7% and 10.1%, respectively. The non-mining population was 7.6% and 22.9%, respectively. The decrease in population in the mining region was significantly smaller than in the non-mining region, indicating that chlorophyll fluorescence was differently affected by Pb stress. Under Pb stress, the net photosynthetic rate of the mined and non-milled populations was significantly positively correlated with Fv/Fm, reaching a correlation coefficient of 0.951 and 0.965, respectively. Under Pb stress, the primary photoconversion efficiency of PSII decreased greatly, and the potential active center of PSII was seriously damaged.

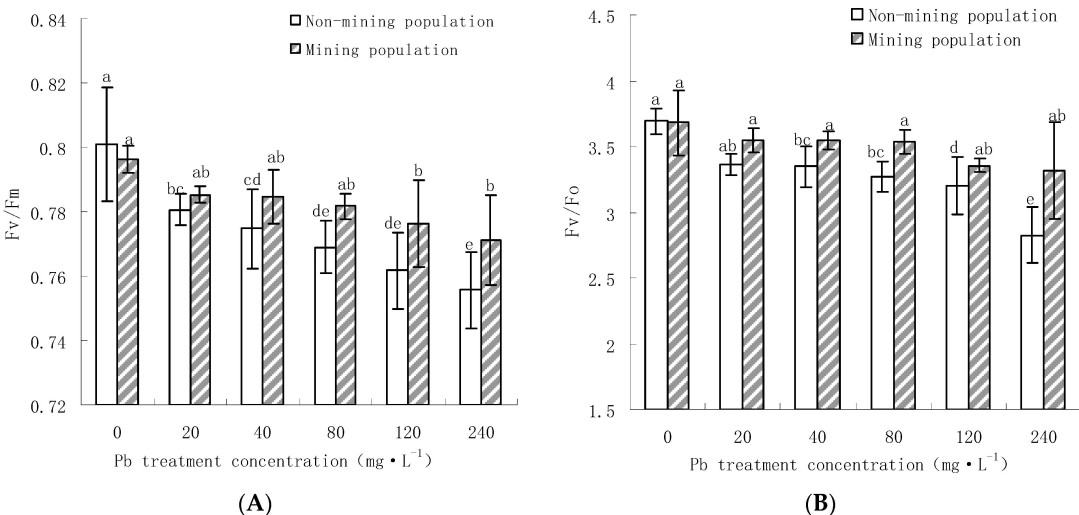

**Figure 4.** Effects of different treatments on the PSII maximal photochemical efficiency Fv/Fm and the ratio of variable fluorescence to initial fluorescence Fv/F$_0$ of two *Miscanthus floridulus* populations under different Pb treatment ((**A**) is Fv/Fm, (**B**) is Fv/F$_0$). Note: Error bars indicate standard deviation; Different letters in the same group indicate significant difference at $p < 0.05$ according to Duncan's multiple range tests; the same below.

### 3.3.2. Effects of Pb on the qP and NPQ of the Leaves of *M. floridulus*

The value of photochemical quenching coefficient qP reflects the redox state of the primary electron acceptor QA of PSII and the opening degree of the reaction center of PSII. The higher the value, the higher the electron transfer activity of PSII [41]. As can be seen in Figure 5A, the qP values of the two populations of *M. floridulus* showed a downward trend under Pb stress. Under mild stress (20 mg·L$^{-1}$ and 40 mg·L$^{-1}$), qP of the two populations decreased slightly compared with the control ($p > 0.05$). As the Pb stress concentration increased, the qP values of the two populations decreased significantly. When Pb concentration was 240 mg·L$^{-1}$, qP values of the two populations decreased by: the population in the mining area (13.8%) and the population in non-mining area (32.2%) showed a relatively large decrease in the non-mining area, indicating that the electron transport activity of PSII reaction center in leaves of the non-mining population of *M. floridulus* was greatly affected by Pb stress.

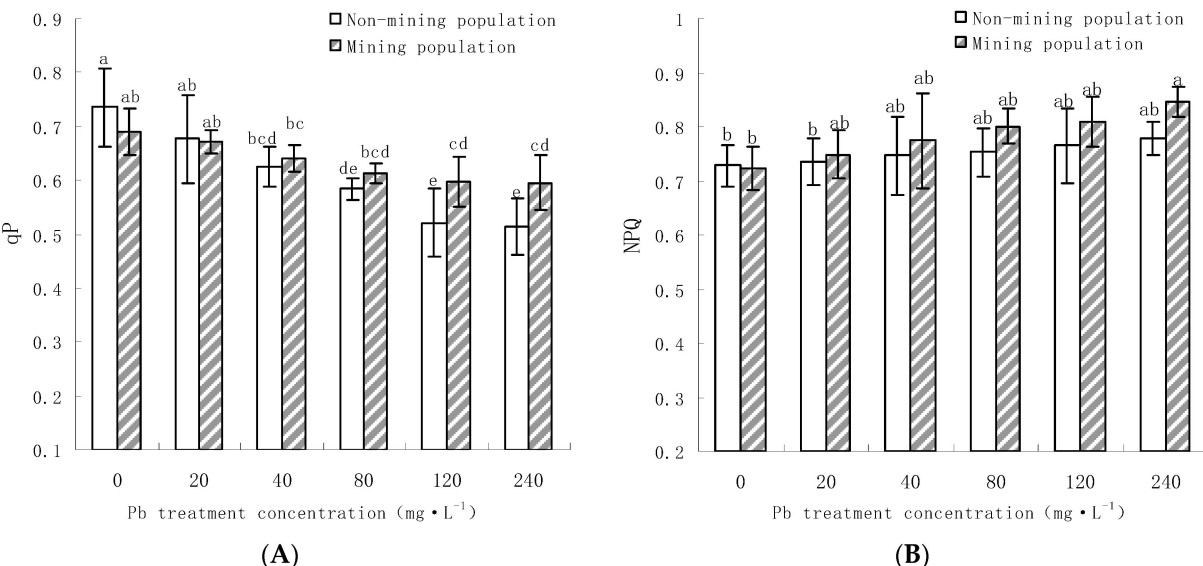

**(A)**                 **(B)**

**Figure 5.** The photochemical quench (qP) and non-photochemical quench (NPQ) of two *Miscanthus floridulus* populations under different Pb treatment ((**A**) is qP, (**B**) is NPQ). Note: Error bars indicate standard deviation; Different letters in the same group indicate significant difference at $p < 0.05$ according to Duncan's multiple range tests; the same below.

The non-photochemical quenching coefficient NPQ reflects the degree of loss of light energy absorbed by blades in the form of heat dissipation. Heat dissipation is an important mechanism to protect PSII from damage caused by photoinhibition [42]. The results show that the NPQ values of the two populations increased under Pb stress. Compared with the control, the NPQ values of the mining population increased under Pb stress (80 mg·L$^{-1}$, 120 mg·L$^{-1}$, and 240 mg·L$^{-1}$) by 11.2%, 12.5%, and 17.8%, respectively. The non-mining population was 2.8%, 4.2%, and 6.7%, respectively. The increase in NPQ value in mining population was higher than that in non-mining population (Figure 5B), indicating that compared with non-mining population with weak resistance, the highly resistant mining population had higher openness of PSII reaction center and stronger electron transfer and heat dissipation ability under Pb stress.

### 3.3.3. Effects of Pb on the Leaf of *M. floridulus* on φPSII and ETR

With the diameter of PSII and ETR reflecting the actual photochemical reaction efficiency and electron transfer rate with partial closure of PSII reaction center under environmental stress [43]. Figure 6A,B show that: with the increase in Pb stress concentration, the diameter of PSII and ETR of the two populations of *M. floridulus* decreased, but the decreas-

ing amplitude of the two populations was different. Compared with the control, under the Pb concentration (80 mg·L$^{-1}$, 120 mg·L$^{-1}$, and 240 mg·L$^{-1}$) stress, mining population $\phi$PSII and ETR decreased 17.8%, 21.9%, 24.1% and 15.5%, 18.1%, 20.1%, respectively, The non-mining population $\phi$PSII and ETR decreased by 31.1%, 40.5%, 52.2% and 22.2%, 31.1%, 48.3%, respectively. It shows that the magnitude of the reduction was much smaller for the mining population than for the non-mining population, and the ETR was much smaller for the mining population. The variance analysis showed that under higher concentration Pb stress (120 mg·L$^{-1}$ and 240 mg·L$^{-1}$), the difference between the mining population and the other treatments was not significant ($p > 0.05$), whereas the difference between the non-mining population and the different treatments was significant ($p < 0.05$).

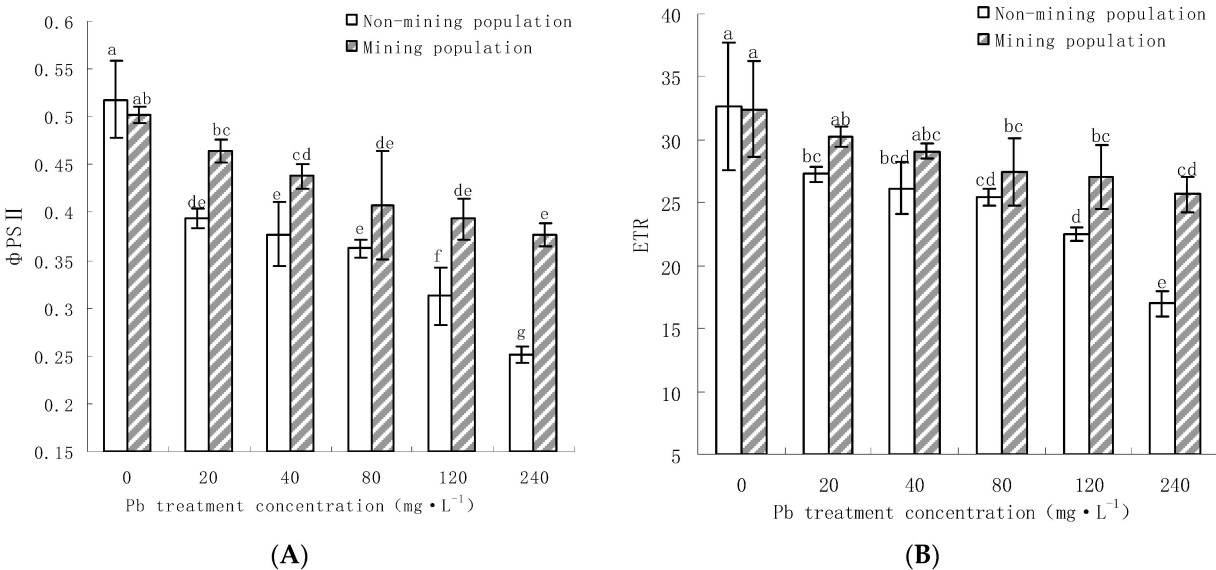

(**A**)                                                           (**B**)

**Figure 6.** The PSII actual photochemical efficiency ($\Phi$PSII) and the acyclic electron transfer rate (ETR) of two *Miscanthus floridulus* populations under different Pb treatment ((**A**) is $\Phi$PSII, (**B**) is ETR). Note: Error bars indicate standard deviation; Different letters in the same group indicate significant difference at $p < 0.05$ according to Duncan's multiple range tests; the same below.

### 3.4. Effects of Pb on Photosynthetic Pigment Content in Leaves of M. floridulus

The content of photosynthetic pigments will affect the absorption and transfer of light energy by chlorophyll, as well as the amount of ATP and NADPH synthesized by the allocation and conversion between PSII and PSI [44], thus affecting photosynthesis. In stressed environments, plants usually show decreased photosynthesis and photosynthetic pigment content after sensing stress [45,46]. As shown in Table 3, the effect of Pb on the photosynthetic pigment content in the leaves of *M. floridulus* varied with different concentrations and populations of Pb. The contents of Chl-a, Chl-b, and carotenoid were increased compared with the control under low Pb concentration (20 mg·L$^{-1}$ and 40 mg·L$^{-1}$) stress. However, as the concentration of Pb increased in the non-mining areas, Chl-a, Chl-b, and Carotenoid rapidly decreased, and the trend was essentially the same for each treatment. Chl a/b increased with Pb concentration for both populations. Under 120 mg·L$^{-1}$ and 240 mg·L$^{-1}$ Pb stress, the content of Chl-a and Chl-b in the mining population was significantly higher than that in the non-mining population ($p < 0.05$).

**Table 3.** Chlorophyll contents (mean ± SD) in *M. floridulus* leaves under different Pb treatment.

| Population | Pb Treatment (mg·L$^{-1}$) | Chl-a Content (mg·g$^{-1}$ FW) | Chl-b Content (mg·g$^{-1}$ FW) | Chl a/b Value (mg·g$^{-1}$) | Car Content (mg·g$^{-1}$ FW) |
|---|---|---|---|---|---|
| Non-mining population | 0 | 1.31 ± 0.010 a | 0.75 ± 0.048 a | 1.74 ± 0.155 d | 0.43 ± 0.005 a |
| | 20 | 1.25 ± 0.013 ab | 0.68 ± 0.055 a | 1.84 ± 0.091 cd | 0.41 ± 0.006 a |
| | 40 | 1.23 ± 0.032 b | 0.52 ± 0.102 b | 2.36 ± 0.472 bcd | 0.39 ± 0.029 b |
| | 80 | 1.11 ± 0.015 c | 0.43 ± 0.041 b | 2.58 ± 0.185 bc | 0.39 ± 0.025 b |
| | 120 | 1.01 ± 0.071 d | 0.29 ± 0.081 c | 3.48 ± 0.523 a | 0.34 ± 0.026 c |
| | 240 | 0.76 ± 0.017 e | 0.25 ± 0.006 c | 3.04 ± 0.663 ab | 0.31 ± 0.033 c |
| Mining population | 0 | 1.26 ± 0.026 a | 0.69 ± 0.009 a | 1.82 ± 0.319 c | 0.39 ± 0.015 ab |
| | 20 | 1.29 ± 0.011 a | 0.72 ± 0.003 a | 1.78 ± 0.061 c | 0.41 ± 0.006 a |
| | 40 | 1.28 ± 0.010 a | 0.75 ± 0.018 a | 1.71 ± 0.063 c | 0.40 ± 0.006 ab |
| | 80 | 1.20 ± 0.020 b | 0.57 ± 0.044 b | 2.11 ± 0.179 bc | 0.38 ± 0.011 ab |
| | 120 | 1.18 ± 0.011 b | 0.51 ± 0.081 bc | 2.31 ± 0.603 ab | 0.36 ± 0.044 ab |
| | 240 | 1.15 ± 0.008 c | 0.45 ± 0.005 c | 2.56 ± 0.075 a | 0.36 ± 0.006 b |

Note: Data in the table are means ± SD ($n = 3$), different letters in the same vertical column indicate significant difference according to SSR (New Complex Range Method) test ($p < 0.05$), the same below.

### 3.5. Correlation between Photosynthetic Physiological Indexes and Pb Concentration

It can be seen from Table 4 that Pn, Gs, Ci, Tr, Fv/Fm, Fv/F$_0$, qP were significantly negatively correlated with Pb treatment concentration among the photosynthetic physiological indexes of the non-mining population of *M. floridulus* ($p < 0.01$). There was no significant correlation between the NPQ and Pb concentration. Among the other indicators, Pn and Gs, Ci, Tr, Fv/Fm, and ETR were significantly positively correlated ($p < 0.01$). Chl-a was positively correlated with Pn, Gs, Ci, Tr, Fv/Fm, Fv/F$_0$, and qP ($p < 0.01$).

**Table 4.** Correlation matrix between photosynthetic physiological factors and Pb concentration of *M. floridulus*.

| Type | Item | Pb | Biomass | Pn | Gs | Ci | Tr | Fv/Fm | Fv/F$_0$ | qP | NPQ | ETR | Chla |
|---|---|---|---|---|---|---|---|---|---|---|---|---|---|
| Non-mining population | Pb | 1 | | | | | | | | | | | |
| | Biomass | −0.868 ** | 1 | | | | | | | | | | |
| | Pn | −0.818 ** | 0.936 ** | 1 | | | | | | | | | |
| | Gs | −0.777 ** | 0.916 ** | 0.897 ** | 1 | | | | | | | | |
| | Ci | −0.820 ** | 0.921 ** | 0.900 ** | 0.839 ** | 1 | | | | | | | |
| | Tr | −0.779 ** | 0.917 ** | 0.980 ** | 0.877 ** | 0.865 ** | 1 | | | | | | |
| | Fv/Fm | −0.711 ** | 0.890 ** | 0.778 ** | 0.812 ** | 0.750 ** | 0.774 ** | 1 | | | | | |
| | Fv/F$_0$ | −0.846 ** | 0.841 ** | 0.873 ** | 0.788 ** | 0.802 ** | 0.811 ** | 0.701 ** | 1 | | | | |
| | qP | −0.750 ** | 0.862 ** | 0.773 ** | 0.786 ** | 0.863 ** | 0.702 ** | 0.728 ** | 0.736 ** | 1 | | | |
| | NPQ | 0.352 | −0.397 | −0.439 | −0.333 | −0.439 | −0.356 | −0.264 | −0.449 | −0.450 | 1 | | |
| | ETR | −0.885 ** | 0.893 ** | 0.813 ** | 0.839 ** | 0.801 ** | 0.770 ** | 0.816 ** | 0.819 ** | 0.857 ** | −0.290 | 1 | |
| | Chla | −0.986 ** | 0.877 ** | 0.819 ** | 0.773 ** | 0.832 ** | 0.775 ** | 0.750 ** | 0.847 ** | 0.759 ** | −0.375 | 0.878 ** | 1 |
| Mining population | Pb | 1 | | | | | | | | | | | |
| | Biomass | −0.860 ** | 1 | | | | | | | | | | |
| | Pn | −0.866 ** | 0.856 ** | 1 | | | | | | | | | |
| | Gs | −0.881 ** | 0.891 ** | 0.861 ** | 1 | | | | | | | | |
| | Ci | −0.686 ** | 0.745 ** | 0.699 ** | 0.685 ** | 1 | | | | | | | |
| | Tr | −0.902 ** | 0.944 ** | 0.873 ** | 0.935 ** | 0.716 ** | 1 | | | | | | |
| | Fv/Fm | −0.456 | 0.665 ** | 0.471 * | 0.716 ** | 0.278 | 0.699 ** | 1 | | | | | |
| | Fv/F$_0$ | −0.565 * | 0.545 * | 0.405 | 0.519 * | 0.288 | 0.548 * | 0.680 ** | 1 | | | | |
| | qP | −0.651 ** | 0.705 ** | 0.771 ** | 0.744 ** | 0.491 * | 0.733 ** | 0.320 | 0.103 | 1 | | | |
| | NPQ | 0.653 ** | −0.664 ** | −0.460 | −0.679 ** | −0.261 | −0.582 * | −0.796 ** | −0.604 ** | −0.292 | 1 | | |
| | ETR | −0.684 ** | 0.716 ** | 0.695 ** | 0.616 ** | 0.432 | 0.710 ** | 0.615 ** | 0.617 ** | 0.376 | −0.538 * | 1 | |
| | Chla | −0.861 ** | 0.904 ** | 0.833 ** | 0.795 ** | 0.771 ** | 0.829 ** | 0.524 * | 0.412 | 0.669 ** | −0.588 * | 0.637 ** | 1 |

\* Correlation is significant at the 0.05 level (2-tailed); ** Correlation is significant at the 0.01 level (2-tailed).

It can be seen from Table 4 that Pn, Gs, Ci, Tr, qP, ETR, Chl-a, and Pb treatment concentration of the mining population of *M. floridulus* were significantly negatively correlated ($p < 0.01$). There was a significant positive correlation between NPQ and Pb concentration ($p < 0.01$). Among the remaining indicators, Pn and Gs, Ci, Tr, and ETR were positively correlated ($p < 0.01$). Chl-a was positively correlated with Pn, Gs, Ci, Tr, and qP ($p < 0.01$).

## 4. Discussion

(1) Specific leaf area (SLA) is one of the important traits of plant leaves. Since SLA is closely related to the growth and survival strategies of plants, it can reflect the adaptation characteristics of plants to different habitats, and is the preferred index of plants in comparative ecological studies [47,48]. The larger the specific leaf area, the larger the leaf area per unit dry weight, and the thinner the leaf. A relatively large leaf area is conducive to capturing more light energy, which increases the possibility of faster growth of plants [49,50]. Studies by Penning et al. [51] also showed that there is a proportional relationship between the maximum photosynthetic rate of plant leaves and the specific leaf area when the nutrient supply is sufficient. In this experiment, with the increase in Pb stress concentration, the specific leaf area (SLA) of both populations decreased, but the magnitude of the decline of the two populations was different. Compared with the control, the specific leaf area (SLA) of mining and non-mining populations decreased to 84.8% and 75.9% and 74.1% and 58.5%, respectively, under the Pb concentration (120 mg·L$^{-1}$ and 240 mg·L$^{-1}$) stress. The results showed that the specific leaf area (SLA) of mining population decreased greatly less than that of non-mining population.

Root/shoot ratio is the final comprehensive effect of many basic change processes and self-adaptation and self-regulation in plants, and so, it can be regarded as the basis of plant structure and function [52]. Compared to plants in the non-mining population, plants in the mining population have a further distribution of above-ground biomass, that is, seedlings in weak light conditions use more energy for the growth of various organs in the above-ground part of the plant, thus obtaining additional light energy for photosynthesis. In conclusion, from the external morphology, the plant growth status of the mining area population was better, especially under the stress of high concentration of Pb (120 mg·L$^{-1}$ and 240 mg·L$^{-1}$).

(2) Chlorophyll (chlorophyll a, chlorophyll b) and carotenoid (carotene and lutein) are the pigments that absorb light energy in the process of plant photosynthesis, among which chlorophyll is the main material that absorbs light energy and directly affects the light energy utilization in vegetation photosynthesis [53,54]. Pb pollution decreased chlorophyll content, Chl a/b ratio and antioxidant enzyme activity, and increased MDA content in tobacco leaves [55]. Chlorophyll fluorescence is commonly used to evaluate the efficiency of photosynthesis and the impact of environmental stress on them [56,57]; the results of this experiment showed that Pb pollution in soil could significantly reduce Fv/F$_0$, Fv/Fm, ETR, and ФPSII, while qP and NPQ also showed an obvious downward trend (Figures 4–6). In this experiment, Pb significantly reduced the leaf chlorophyll content of the non-mining population of *M. floridulus*, and the chlorophyll content of the mining population increased under low Pb concentration (20 mg·L$^{-1}$ and 40 mg·L$^{-1}$) treatment, but decreased significantly under high Pb (120 mg·L$^{-1}$ and 240 mg·L$^{-1}$) treatment. However, chlorophyll levels were still significantly higher than in the non-mining population. This was likely because some resistance mechanisms were developed in the *M. floridulus* population in mining areas to reduce the damage caused by Pb to chlorophyll.

(3) Farquhar et al. [58] suggested that the reduction in Pn is caused by both gastric and non-gastric restriction. When Pn and Ci vary in the same direction, both decrease at the same time, Ls increase, and Pn's decline is mainly caused by Gs; otherwise, Pn decline could be attributed to the decreased carboxylation ability of mesocytes. In addition, chloroplast structure is closely related to photosynthesis, and the more the number of chloroplast grana and grana lamellae, the denser the arrangement of grana lamellae and the stronger the photosynthetic capacity [59]. In the present study, the decrease in Pn in the non-mining population of *M. floridulus* under mild Pb stress was accompanied by a significant decrease in Ci and a large increase in Ls, indicating that gastric restriction was the main factor responsible for the decrease in Pn in the non-mining population of *M. floridulus* under mild Pb stress. However, with further reduction in Pn under moderate and severe Pb stress, Ci and Ls showed minor changes compared to the control case. These results suggest that the reduction in Pn in the non-mining population under moderate and severe Pb stress was

mainly caused by the reduced photosynthetic activity of mesophyll cells, with non-stomatal restriction playing a major role. In the mining population, Pn decreased only at moderate and severe Pb stresses, while Ci decreased and Ls increased, indicating that the decrease in Pn under Pb stresses was mainly due to the gastral restriction.

(4) Photosynthesis is an extremely important metabolic process in plants, and leaf chlorophyll fluorescence is closely related to various reaction processes in photosynthesis. The influence of any environmental factors on photosynthesis can be reflected through leaf chlorophyll fluorescence dynamics; thus, chlorophyll fluorescence is an ideal internal probe to detect the dynamic changes of plant photosynthesis [60]. In photosynthesis, the value of Fv/Fm is commonly used to indicate the environmental stress on the plant. Fv/Fm measured after dark adaptation reflects the quantum yield when all PSII reaction centers are in the open state, namely the potential maximum quantum yield of PSII [61]. In normal physiological state, Fv/Fm is a relatively stable value, and the value of Fv/Fm is generally around 0.8 [62]. In drought stress [63], salt stress [64], and light stress [65], it was found that the value of Fv/Fm decreased with the increase in stress degree. The photosynthetic inhibition indicated by Fv/Fm to be lower than 0.8 may have been due to photoprotection or damage of the PSII reaction center [66]. In this experiment, the Fv/Fm of the mining population of *M. floridulus* was around 0.8 and the correlation with Pb was low. However, at high Pb stress, the Fv/Fm of the non-mining population of *M. floridulus* dropped to about 0.7, indicating that the effect of Pb stress on photosynthesis was much greater in the non-mining population than in the mining population.

## 5. Conclusions

(1) Under Pb stress, the net photosynthetic rate (Pn), transpiration rate €, stomatal conductance (Gs), intercellular carbon dioxide concentration (Ci), and chlorophyll content (Chl) of the leaves of the two populations of *M. floridulus* decreased in different amplitude. With the increase in Pb stress, the net photosynthetic rate (Pn) of the mining population decreased slowly, and the plant growth and biomass were less affected.

(2) Stomatal restriction was the main factor in the reduction in Pn in the non-mining population of *M. floridulus* under mild Pb stress and non-stomatal restriction was the main factor in the reduction in Pn under moderate and severe Pb stress.

(3) The maximum photochemical efficiency (Fv/Fm), potential activity (Fv/Fo), and effective photochemical efficiency (Fv'/Fm') of PSII reaction center decreased under Pb stress. On the whole, the decrease in population in the mining area was small, and the utilization of light energy and the potential activity of PSII were strong. The changes of photochemical quenching coefficient (qP) and non-photochemical quenching coefficient (NPQ) of PSII showed that qP value decreased and NPQ value increased in the two populations under Pb stress. Overall, the population in the resistant mining region had a low qP reduction and a large NPQ increase. With the increase in Pb stress concentration, the actual photochemical reaction efficiency (PSII) and electron transfer rate (ETR) on the mining population did not change much, and the non-mining population decreased significantly. This indicates that the electron transfer activity of PSII reaction center was less affected and the damage degree of photosynthetic mechanism was low. The results show that the mining population of *M. floridulus* had a strong tolerance to Pb, which is suitable for the pioneer species of Gramineae in the vegetation restoration construction in the metal mining area.

**Author Contributions:** Project administration and writing—review and editing, J.Q. (Jianqiao Qin); conceptualization, X.J. and H.Z.; formal analysis, J.Q. (Jianhua Qin); methodology, M.D. and X.C.; investigation, H.L. All authors have read and agreed to the published version of the manuscript.

**Funding:** This work was supported by the Guangdong Provincial Key Laboratory of Environmental Health and Land Resource (project number: 2020B121201014), the Special Project of Key Areas of Colleges and Universities in Guangdong Province (Science and Technology Promoting Rural Revitalization) "Research and Development of Key Technologies for Resource Utilization of Manure from Large-Scale

Livestock and Poultry Breeding in Rural Areas of Western Guangdong" (No.:2021ZDZX4023), the Innovation Team Project of Colleges and Universities in Guangdong Province (2021KCXTD055), and 2021 Guangdong Provincial Department of Education "Quality Engineering" College Student Social Practice Teaching Base Construction Project (Yue Jiao Gao Han [2021] No. 29).

**Institutional Review Board Statement:** Not applicable.

**Informed Consent Statement:** Not applicable.

**Data Availability Statement:** The data presented in this study are available on request from the corresponding author.

**Acknowledgments:** We acknowledge the Guangdong Provincial Key Laboratory of Environmental Health and Land Resource; Special Project of Key Areas of Colleges and Universities in Guangdong Province (Science and Technology Promoting Rural Revitalization) "Research and Development of Key Technologies for Resource Utilization of Manure from Large-Scale Livestock and Poultry Breeding in Rural Areas of Western Guangdong" and the Innovation Team Project of Colleges and Universities in Guangdong Province.

**Conflicts of Interest:** The authors declare no conflict of interest.

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
