# Peer review of "Effects of Lead Pollution on Photosynthetic Characteristics and Chlorophyll Fluorescence Parameters of Different Populations of Miscanthus floridulus"

_processes, doi:10.3390/pr11051562_

Round 1

Reviewer 1 Report

article presents a very interesting topical study.

I have a number of observations:

1. I have a number of observations - but in the work only Pb is studied. please change the title to reflect the specifics of the paper

2. `please have the purpose of the paper (line 90) identified as a separate paragraph so that it can be identified as easily as possible

3. in my opinion, the experimental part and the way of determining the studied parameters should be made more explicit, but if a series of bibliographical references are provided in which these elements are presented more extensively, I say it is ok.

4. line 188 - P<0.05.  I think it should be written p<0.05 (line 191)

5. lines 204-205 - I think it's a mistake generated by the pdf conversion program

6. the explanations of the results obtained are very brief. please expand them.

7. approximately to each figure we have (a) and (b) but no explanation to the figure title

8. line 402 - there are two points - you need one

9. there are many bibliographical references (68 references) which denotes that the author has studied the field. but there are a number of very old references - Reference 1 1998, reference 2 1989, reference 4 1985. please bring it to at least 10 years old

Reviewer 2 Report

#Comments (CHAF-D-23-00242)

The authors studied the effects of lead pollution on the photosynthetic characteristics and chlorophyll fluorescence parameters of different populations of Miscanthus floridulus. First, the authors germinated the seeds in nutrient-rich soil, then pre-cultured the plants. Pb with CK (what do you mean by CK!!), 20, - 240; Pb was added to the nutrient solution. but I wonder how you can examine the bioavailability of Pb when you have used Pb(NO3)2, knowing that you are adding EDTA as a chelating agent at a high concentration of 12.5??. In addition to the high probability of precipitation with the components of the medium, especially KH2PO4 ???

Then, the authors determined some physiological parameters: plant morphology, gas exchange parameters, chlorophyll fluorescence parameters and photosynthetic pigments content. Without addressing the biochemical and enzymatic mechanisms involved in the management of oxidative stress in plants, I invite the authors to refer to this review: https://doi.org/10.1007/s10811-021-02668-w

- Please check the text it has words written in chinos (和!!!)

- Write in italics all names of strain

The following shortcomings are evident in the submitted manuscript.

# Abstract:

ü    The abstract need to be modified which abide to the rules of the journal's regulation. Further, the major results obtained in this work need to represent in the abstract.

# Introduction:

The objective of the study is not clear! the introduction's framework needs to be reconsidered. And the context and purpose of the research are not clearly defined.

ü    What is the gap to cover?

ü    Gap analysis/need for study must be highlighted in the introduction part prior to the objectives.

ü    Please add references at the end of all sentences for informational purposes

ü    Please rewrite the introduction. we invite you to consult the following articles dealing with this subject https://doi.org/10.1007/s42398-020-00150-w;

ü    Please explain why the authors chose M. floridulus and targeted Pb metals at this particular concentration.

#Results and discussion

ü    Please improve the analysis and interpretation of results.

ü    Please compare the experimental results with other references.

ü    Captions of the figure and tables must be with complete information,

#Conclusion

Please rephrase it.

#References

Please withdraw the reference from an unindexed journal or Not Validated

Reviewer 3 Report

Authors need to carefully counter-check the language and typo errors before submission. 

Round 2

Reviewer 3 Report

Authors have corrected the manuscript. But there are some issues which were overlooked by the authors. I urged them to address those for finalizing the MS. 

1. Add some quantitative data like percentage change or probability to support your finding in abstract.

2. Line 171-72: Delete third penultimate. Just mention ".....fully grown leaf was choose......" instead.

3. Line 190-204 (Section 2.4, 2.5) :  Still not clear. Authors are advised to carefully revisit these and revised these sections.

4. Line 254: Pls provide full name of SSR (regression sum of squares) in legend as well.

5. References are not uniform, still need revision specially journal names. 

OK
